# Constituents of the Stem Bark of *Trichilia monadelpha* (Thonn.) J. J. De Wilde (Meliaceae) and Their Antibacterial and Antiplasmodial Activities

**DOI:** 10.3390/metabo13020298

**Published:** 2023-02-17

**Authors:** Arnauld Kenfack Djoumessi, Raymond Ngansop Nono, Beate Neumann, Hans-Georg Stammler, Gabin Thierry Mbahbou Bitchagno, Noella Molisa Efange, Celine Nguefeu Nkenfou, Lawrence Ayong, Bruno Ndjakou Lenta, Norbert Sewald, Pépin Nkeng-Efouet-Alango, Jean Rodolphe Chouna

**Affiliations:** 1Department of Chemistry, Faculty of Science, University of Dschang, Dschang P.O. Box 67, Cameroon; 2Department of Chemistry, Inorganic and Structural Chemistry, Bielefeld University, P.O. Box 100131, 33501 Bielefeld, Germany; 3Centre Pasteur du Cameroun, Yaounde P.O. Box 1274, Cameroon; 4Department of Biology, Higher Teacher Training College, University of Yaoundé 1, Yaoundé P.O. Box 47, Cameroon; 5Molecular Biology Center, Yaoundé P.O. Box 14475, Cameroon; 6Department of Chemistry, Higher Teacher Training College, University of Yaoundé 1, Yaoundé P.O. Box 47, Cameroon; 7Department of Chemistry, Organic and Bioorganic Chemistry-OC3, Bielefeld University, P.O. Box 100131, 33501 Bielefeld, Germany

**Keywords:** Meliaceae, *Trichilia monadelpha*, limonoids, antibacterial, antiplasmodial

## Abstract

The chemical investigation of the EtOH extract from the stem bark of *Trichilia monadelpha* (Thonn.) J. J. De Wilde afforded two new limonoids (**1** and **2**): 24-acetoxy-21,25-dihydroxy-21,23-epoxytirucall-7-en-3-one (**1**) and (6*R*)-1-*O*-deacetylkhayanolide E (**2**), together with eleven known compounds (**3**–**13**), including additional limonoids, flavonoids, triterpenoids, steroids, and fatty acid. Their structures were determined using 1D- and 2D-NMR experiments, ESI mass spectrometry, and single crystal X-ray diffraction analysis. The antibacterial and antiplasmodial activities of the extracts, sub-extracts, fractions, and some of the isolated compounds were evaluated in known pathogenic strains, including *Staphylococcus aureus* and *Plasmodium falciparum*. Fraction E (*n*-Hex/EtOAc 30:70, *v*/*v*) showed significant activity against *S*. *aureus* ATCC 25923 with a MIC value of 3.90 µg/mL, while one of its constituents (epicatechin (**9**)) exhibited significant activity with MIC values of 7.80 µg/mL. Interestingly, grandifotane A (**6**) (IC_50_ = 1.37 µM) and khayanolide D (**5**) (IC_50_ = 1.68 µM) were highly active against the chloroquine-sensitive/sulfadoxine-resistant *plasmodium falciparum* 3D7 strain, unlike their corresponding plant extract and fractions.

## 1. Introduction

*Trichilia* represents the largest genus in the Meliaceae family, comprising approximately 70 species mainly distributed across the tropical regions of Africa and America [1]. In Cameroon, *T*. *monadelpha* was previously known as *T. heudelotii* [2]. The extract of its stem barks is used in traditional medicine for the treatment of central nervous system (CNS) conditions such as epilepsy, depression, psychosis, inflammation, and pain [3,4]. It is also used to treat skin ulcers, syphilis, coughs, gastrointestinal pains, gonorrhea, and rheumatism [2,5,6,7]. A decoction of its leaves is taken to treat heart problems [8,9]. Previous phytochemical screenings of *T*. *monadelpha* have indicated the presence of limonoids [10,11,12,13], alkaloids [14], terpenes [13,15], tannins [9], flavonoids [4], cardiac glycosides [16], steroids [13,17], and saponins [9,16]. However, limonoids (monadelphin A, monadelphin B) [13], diterpenes (nimbiol, isopimarinol, 7-ketoferruginol, 12β-hydroxysandaracopimar-15-ene) [15], sesquiterpenes (trichins A and B) [13], coumarin (scopoletin) [13], phenolic acids (ellagic acid, protocatechuic acid, 2-propionoxy-β-resorcylic acid, 4-hydroxybenzoic acid, 2-methylprotocatechuic acid), and the alkaloid coixol [13,15] were also isolated. Some of these isolates are reported to possess antimalarial [18], anti-inflammatory, antitumor [16,19], antimicrobial [13,15], analgesic [4,20], antianaphylactic [4,21], and antioxidant activities [4,14,16,19]. As part of our ongoing search for bioactive compounds from Cameroonian medicinal plants [22,23,24], we report thirteen secondary metabolites, including two new limonoids (**1**, **2**) obtained from the ethanol extract of *T. monadelpha* stem bark, which we have assessed for their antibacterial and antiplasmodial activities against known pathogenic strains. 

## 2. Materials and Methods

### 2.1. General Experimental Procedures

The optical rotations were determined with a Jasco DIP-3600 digital polarimeter (Jasco, Tokyo, Japan) using a 10 mm cell. The high-resolution mass spectra were recorded using a Micromass-Q-TOF-MS (Waters, Milford, MA, USA). For the DMSO-*d_6_*, MeOD, CDCl_3_, and Acetone-*d*_6_, the ^1^H NMR and ^13^C NMR spectra were recorded using Bruker DRX (^1^H NMR, 500 MHz and ^13^C NMR, 125 MHz) and Bruker Avance 600 (^1^H NMR, 600 MHz, and ^13^C NMR, 150 MHz) spectrometers (Bruker, Rheinstetten, Germany), respectively. X-ray crystallography data were collected with a Rigaku Supernova diffractometer using Cu Kα (λ = 1.54184 Å) radiation. Column chromatography (CC) readings were carried out using silica gel (63–200 μm, Merck, Darmstadt, Germany), and Sephadex LH-20. A TLC analysis was performed using percolated aluminum plates backed with silica gel 60 F_254_ sheets. The TLC plate was visualized under UV light (254 and 365 nm), sprayed with H_2_SO_4_ (10%), and then heated.

### 2.2. Plant Material 

The stem bark of *Trichilia monadelpha* was collected in Mawa in the Noun sub-division of the West Region in Cameroon in September 2016, and subsequently identified by comparing it to a voucher specimen (66909/HNC) at the National Herbarium of Cameroon (NHC), Yaoundé.

### 2.3. Extraction and Purification

The fresh stem bark was air-dried, powdered (2.1 kg), and then underwent extraction by macerating it in EtOH (9 L) at room temperature for 72 h. The filtrate was concentrated under reduced pressure to give 155.0 g of crude extract. Part of this extract (150.0 g) was then partitioned with EtOAc (68.1 g) and *n*-BuOH (10.4 g). A part of the EtOAc sub-extract (65.0 g) was subjected to column chromatography using silica gel, and then eluted with *n*-Hex/EtOAc (93:07 to 30:70, *v*/*v*) and EtOAc/MeOH (95:05, *v*/*v*) to give six major fractions, indexed from A to F. Fraction A (*n*-Hex/EtOAc 93:7, 6.1 g) was chromatographed using a silica gel column with a gradient elution of *n*-Hex/EtOAc (99:1 to 93:7, *v*/*v*) to yield stigmastane-3,6-dione **10** (5 mg). Fraction B (*n*-Hex/EtOAc 85:15, 3.2 g) was separated using silica gel column chromatography and then eluted with *n*-Hex/EtOAc (93:7 to 85:15, *v*/*v*) to afford a mixture of *β*-sitosterol (**12a**), stigmasterol (**12b**) (50 mg), and tetracosanoic acid **11** (10 mg). Fraction C (*n*-Hex/EtOAc 75:25, 4.3 g) was subjected to silica gel column chromatography and eluted with *n*-Hex/EtOAc (85:15 to 75:25, *v*/*v*) to afford 24-acetoxy-21,25-dihydroxy-21,23-epoxytirucall-7-en-3-one **1** (7 mg) and (+)-21*R**,23*R**-epoxy-21*α*-methoxy-24,25-dihydroxyapotirucall-7-en-3-one **4** (5.2 mg). Separation of fraction D (*n*-Hex/EtOAc 50:50, 12.1 g) was achieved with CC using silica gel and eluted with *n*-Hex/EtOAc (75:25 to 50:50, *v*/*v*) to yield betulinic acid **8** (10 mg), betulin **7** (15 mg), melianodiol **3** (350 mg), and grandifotane A **6** (150 mg). Fraction E (*n*-Hex/EtOAc 30:70, 15.1 g) was chromatographed using a silica gel column chromatography and eluted with *n*-Hex/EtOAc (50:50 to 25:50, *v*/*v*) to provide khayanolide D **5** (15 mg), (6*R*)-1-*O*-deacetylkhayanolide E **2** (12 mg), and epicatechin **9** (100 mg). Fraction F (EtOAc/MeOH 95:05, 18 g) was separated with silica gel CC, using EtOAc/MeOH (100:0 to 70:30, *v*/*v*) as eluent to afford *β*-sitosterol 3-O*-β*-D-glucopyranoside **13** (400 mg).

### 2.4. X-ray Crystallography

X-ray crystallography data of (6*R*)-1-*O*-deacetylkhayanolide E (**2**): A colorless crystal was obtained from CH_2_Cl_2_–MeOH (3:7). Cell parameters: orthorhombic, space group P2_1_2_1_2_1_ (no. 19), *a* = 7.61550(10) Å, *b* = 11.09580(10) Å, *c* = 28.0745(2) Å, *V* = 2372.30(4) Å^3^, *Z* = 4, *T* = 100.0 (1) K, μ(Cu Kα) = 0.925 mm^−1^, *Dcalc* = 1.446 g/cm^3^, 85,714 reflections measured (6.3° ≤ 2Θ ≤ 152.8°), 4957 unique (*R*_int_ = 0.0358, *R*_sigma_ = 0.0103) which were used in all calculations. The final *R*_1_ was 0.0329 for 4891 reflections with *I > 2σ(I)*, and *wR*_2_ was 0.0813 for all data, with the Flack parameter −0.02(3). The absolute configuration was determined to be *R* for C2, C4, C5, C6, C9, C10, C14, and C17; and S for C1, C8, C13, and C30. CCDC 2176790 contains the supplementary crystallographic data for this paper. These data can be obtained free of charge from The Cambridge Crystallographic Data Centre via https://www.ccdc.cam.ac.uk/structures/ (accessed on 10 February 2022)

### 2.5. Biological Assays 

#### 2.5.1. In Vitro Antibacterial Activity

Four Gram-negative bacteria (*Salmonella typhi*, *Pseudomonas aeruginosa* NR 48982, *Klebsiella pneumoniae* NR 41388, and *Klebsiella pneumoniae* clinical isolate) and three Gram-positive bacteria (*Staphylococcus aureus* ATCC 25923, *Staphylococcus aureus* ATCC 43300, and *Staphylococcus aureus* clinical isolate) were tested for their susceptibility to extracts and compounds isolated from *T*. *monadelpha*. The minimum inhibitory concentration (MIC) of the samples was evaluated following the broth microdilution method as described by Eloff [25], with slight modifications. Extracts, fractions, sub-fractions, compounds, and the reference drug were dissolved in DMSO. The bacterial suspension prepared as an inoculum was adjusted to a turbidity equivalent to that of a 0.5 McFarland standard to achieve approximately 1.5 × 10^8^ CFU/mL. Ciprofloxacin was used as a positive control. A volume of one hundred microliters of Mueller Hinton Broth (MBH) was added into all wells of the 96-well plate, and 100 µL of the compounds/extracts, fractions, and sub-fractions were introduced to the wells in the first row (A), and then mixed thoroughly. A volume of 100 µL of this sample mixture was removed from the wells of row A to perform a two-fold serial dilution down the rows (B–H). The last 100 µL was discarded. Then, 100 µL of the inoculum was introduced into all the wells. The final volume in each well was 200 µL. Each sample concentration was assayed in triplicate, and each test was performed twice. After an incubation period of 18 h at 37 °C, 20 µL of Alamar Blue was added to each well. The plates were then reincubated for 30 min at 37 °C. A blue color in the well was scored as “no bacterial growth”, and a pink color was scored as “growth occurrence”. MIC values were defined as those concentrations in which a pronounced change in color was noticed (from blue to pink).

#### 2.5.2. In Vitro Antiplasmodial Activity: *Plasmodium falciparum* Culture and Growth Inhibition Assay 

*Plasmodium falciparum* 3D7 (chloroquine-sensitive/sulfadoxine-resistant) strain was maintained in 5% CO_2_ at 37 °C using a modified Trager and Jensen method [26] in fresh O+ human red blood cells at 3% hematocrit in RPMI culture media containing NaHCO_3_ (Gibco, UK) and GlutaMAX supplement. This was supplemented with hypoxanthine (Gibco, Waltham, MA, USA), 25 mM of HEPES (Gibco, Drewton, UK), 0.5% Albumax II (Gibco, Waltham, MA, USA), and 20 µg/mL of gentamicin (Gibco, China). When needed, parasites were synchronized at the ring stage with a sorbitol (5%) treatment and further cultivated for one complete cycle (48 h) prior to the drug activity assays. Compounds dissolved in dimethyl sulfoxide (DMSO) were diluted in RPMI 1640 and mixed with the parasite cultures (1.5% hematocrit and 1% parasitemia, respectively) in 96-well plates to achieve final drug concentration levels of 10 μM for primary screening assays, and 10–0.078 μM for the dose-dependent response assays. The final DMSO concentration per 100 μL culture per well was 0.1%. Artemisinin and chloroquine at 1 µM were used as positive drug controls, while (0.1%) DMSO was used as a negative drug control. Following a 72 h incubation at 37 °C, parasite growth was assessed using a SYBR green I-based DNA quantification assay. Briefly, 80 µL of parasitized erythrocytes were transferred to a dark plate and 40 µL of an SYBR green I-containing lysis buffer (3×) was added to the plate. The plate was incubated in the dark for 30 min and its fluorescence was measured using a Fluoroskan Ascent multi-well plate reader with excitation and emission wavelengths at 485 and 538 nm, respectively. The experiments were performed in triplicate and each one was repeated at least once. The concentrations at which 50% inhibition of growth (IC_50_ values) was obtained were determined using GraphPad Prism 8.0, by plotting the logarithmic sample concentration on the *x*-axis against the percentage of inhibition on the *y*-axis.

## 3. Results and Discussion

### 3.1. Chemical Investigation

The EtOH extract of the stem bark of *T. monadelpha* was separated into four fractions with liquid–liquid partition using EtOAc and n-BuOH, respectively. The EtOAc fraction was subjected to repeated silica gel and Sephadex LH-20 column chromatography to afford thirteen compounds, including 24-acetoxy-21,25-dihydroxy-21,23-epoxytirucall-7-en-3-one (**1**), (6*R*)-1-*O*-deacetylkhayanolide E (**2**), melianodiol (**3**) [27], (+)-21*R**,23*R**-epoxy-21*α*-methoxy-24,25-dihydroxyapotirucall-7-en-3-one (**4**) [28], khayanolide D (**5**) [29], grandifotane A (**6**) [30], betulin (**7**), betulinic acid (**8**) [31], epicatechin (**9**) [32], stigmastane-3,6-dione (**10**) [33], tetracosanoic acid (**11**) [34], the mixture of *β*-sitosterol (**12a**) and stigmasterol (**12b**) [35], and *β*-sitosterol 3-O*-β*-D-glucopyranoside (**13**) [36] (Figure 1).

Compound **1** was a white powder with an ${\left[\alpha \right]}_{D}^{20}$ of −13.5 (*c* 1, MeOH). Its HRESIMS in positive mode showed a sodium adduct ion [M + Na]^+^ at *m*/*z* 553.3491 (calculated as 553.3499 for C_32_H_50_O_6_Na^+^), indicating eight double bond equivalents. The ^1^H NMR spectrum of **1** (Table 1, Appendix A) exhibited signals of eight methyl singlets at *δ*_H_ 0.90 (3H, *s*, H-18), 1.08 (3H, *s*, H-19), 0.93 (3H, *s*, H-26), 1.23 (3H, *s*, H-27), 1.05 (3H, *s*, H-28), 1.15 (3H, *s*, H-29), 1.09 (3H, *s*, H-30), and 2.12 (3H, *s*, H-2′), and an olefinic proton at *δ*_H_ 5.37 (1H, *brs*, H-7). In addition, oxygenated methines were observed at *δ*_H_ 4.34/4.25 (1H, *dt*, *J* = 9.5, 4.8 Hz, H-23); 4.99/4.93 (1H, *d*, *J* = 5.4 Hz, H-24), and a hemi-acetal proton at *δ*_H_ 5.19/5.18 (1H, *d,* 5.3/3.6, H-21). The ^13^C NMR (Appendix A) and DEPT (Appendix A) data indicated 32 carbon resonances sorted into eight methyl groups at *δ*_C_ 22.3 (C-18), 11.7 (C-19), 21.7 (C-26), 25.2 (C-27), 23.6 (C-28), 25.4/25.3 (C-29), 26.4 (C-30), and 19.7/19.6 (OCOCH_3_); one hemi-acetal carbon at *δ*_C_ 101.0/96.7 (C-21); two carbonyl groups at *δ*_C_ 216.8 (C-3) and 170.8 (OCOCH_3_); and two olefinic carbons at *δ*_C_ 117.9 (C-7) and 145.7/145.6 (C-8). All these data were characteristic of limonoids of the tirucallane series, and they were superimposable over those of melianodiol [27,36,37]. The only difference was the additional signals of the acetyl group at *δ*_C_ 19.7/19.6 (C-1′) and 170.8 (C-2′) in compound **1**. The proposed structure was supported by the ^1^H-^1^H COSY (Appendix A) and the HMBC correlations (Figure 2 and Appendix A). Compound **1** was a C-21 (*δ*_C_ 96.7/101.0) epimeric mixture similar to melianodiol, 21-methoxylmelianodiol, and other known tirucallane-types, possessing a similar hemiacetal side chain [27,38,39]. The relative configuration of the asymmetric carbons can be attributed to its structural similarities to melianodiol (**3**) [27]. Therefore, compound **1** was determined to be 24-acetoxy-21,25-dihydroxy-21,23-epoxytirucall-7-en-3-one.

Compound **2** was obtained as a white crystal with an ${\left[\alpha \right]}_{D}^{20}$ of = −2.1 (*c* 1.0, MeOH). The molecular formula C_27_H_32_O_10_Na^+^ was determined from its positive HRESIMS (Appendix A), which exhibited the sodium adduct ion [M + Na]^+^ at *m*/*z* 539.1893 (calculated as 539.1887 for C_27_H_32_O_10_Na^+^). The ^1^H NMR spectrum (Table 1, Appendix A) exhibited signals of three methyl groups at *δ*_H_ 1.19 (3H, *s*), 1.23 (3H, *s*), and 1.44 (3H, *s*); three oxymethine protons at *δ*_H_ 4.40 (1H, *d*, *J* = 10.5 Hz), 4.49 (1H, *d*, *J* = 1.9 Hz), and 5.59 (1H, *s*); three methines at *δ*_H_ 2.44 (1H, *d*, *J* = 9.5 Hz), 2.91 (1H, *d*, *J* = 10.5 Hz), and 3.60 (1H, *d*, *J* = 1.7 Hz); a methoxy group at *δ*_H_ 3.83 (3H, s); and three downfield shifted signals attributed to a 3-substituted furan ring at *δ*_H_ 6.46 (1H, *d*, *J* = 2.0 Hz), 7.53 (1H, *t*, *J* = 1.9 Hz), and 7.52 (*d*, 2.6 Hz). The oxymethine proton at *δ*_H_ 4.49 (*J* = 1.9 Hz, H-6) attached to a carbon adjacent to an ester carbonyl, and again coupled in ^1^H-^1^H COSY with a doublet of proton at *δ*_H_ 3.60 (*J* = 1.9 Hz, H-5). The presence of this moiety and the characteristic oxymethine H-17 at *δ*_H_ 5.59 (1H, s) suggested that **2** was a B,D-ring seco-limonoid. The absence of signals of two tertiary methyls at C-29 and C-30, and the presence of methylene at *δ*_H_ 2.23 (d, *J* = 12.6 Hz, H-29a) and 1.88 (d, *J* = 12.5 Hz, H-29b) in the basic limonoid skeleton, supported the notion that **2** was a phragmalin-type limonoid [40,41].

The ^13^C NMR (Appendix A) spectrum displayed the signals of 27 carbon atoms sorted using both DEPT 135 (Appendix A) and HSQC (Appendix A) into four methyl groups; four methylenes; nine methines, including three furan methines at *δ*_C_ 110.8, 142.1, and 144.0; and then quaternary carbons, including three carbonyls at *δ*_C_ 173.0, 176.5, and 208.5. All these data were similar to those of 1-*O*-deacetylkhayanolide E (**2a**) [40]; the only difference was the coupling constant between protons H-5 and H-6 (*J* = 1.9 Hz instead of *J* = 8.7 Hz) in 1-*O*-deacetylkhayanolide E. This information led to the conclusion of a *cis* orientation for H-5/H-6. The ^1^H-^1^H COSY (Appendix A) and the HMBC (Appendix A) correlations (Figure 2) supported the heptacyclic skeleton of the proposed structure, which was then confirmed using single crystal X-ray diffraction analysis. Compound **2** contained twelve asymmetric carbon atoms; the absolute configurations were obtained with a Flack parameter of −0.02(3) (Figure 3). The C-6 configuration was determined to be *R*, while it was reported to be *S* for the known khayanolide E and 1-*O*-deacetylkhayanolide E (**2a**) [40]. Thus, compound **2** was determined to be (6*R*)-1-*O*-deacetylkhayanolide E, a C-6 epimer of **2a**.

### 3.2. Biological Activity

#### 3.2.1. Antibacterial Activities

The antibacterial activities (Table 2) of the extract, EtOAC and n-BuOH sub-extract, fractions (FA, FB, FC, FC, FD, FE, FF), and some of the isolated compounds (**1**, **3**, **5**, **6**, **7**, **9**) were evaluated against seven bacterial strains, including four Gram-negative bacteria (*Salmonella typhi*, *Pseudomonas aeruginosa* (NR 48982), *Klebsiella pneumoniae* (NR 41388), *Klebsiella pneumoniae* (clinical isolate)), and three Gram-positive bacteria (*Staphylococcus aureus* (ATCC 25923), *Staphylococcus aureus* (ATCC 43300), and *Staphylococcus aureus* (clinical isolate)). Almost all the tested samples showed weak to significant activity with MIC values as low as 3.9, compared to the reference compound ciprofloxacin (MIC: 0.015 to 0.0625 µg/mL), as reported by Kuete and Efferth [42]. The ethanol extract showed good activity against *S. aureus* ATCC 43300, *P. aeruginosa* (NR 48982), and *S*. *typhi* CPC with MIC values of 62.5 µg/mL, 125 µg/mL, and 250 µg/mL, respectively. However, it was almost inactive against the other tested strains. The EtOAc and the *n*-BuOH sub-extract were marginally less active against all tested strains. 

Fraction E showed the most significant activity (MIC values ranging from 3.9 to 250 μg/mL), and the best potency was observed against *S*. *aureus* (ATCC 25923, MIC = 3.9 µg/mL). Compound **9**, isolated from fraction E, showed good activity (MIC = 7.8 µg/mL) against *S*. *aureus* (ATCC 25923), and may be the active principle. Compounds **3** isolated from fraction D exhibited moderate activity against *S*. *aureus* (ATCC 25923) and *K*. *pneumoniae* clinical isolate with a MIC value of 62.5 µg/mL. The other compounds (**1**, **5**, **6**, **7**) were not active on the selected strains. The lack of activity of **1** compared to **3** could be attributed to the 24-OAc group. The activity of compound **3** was in agreement with the result obtained by Biavatti et al. [43] with the melianodiol on *S*. *aureus* (ATCC 6538, MIC = 25 µg/mL). However, its activity on *K. pneumoniae* strains was not yet evaluated. In addition, the activity of compound **9** was in accordance with that of Masika et al. [44], which showed the antibacterial activity of epicatechin against *S*. *aureus* strain (MIC = 250 µg/mL). In general, the observed antibacterial activity of limonoids was very poor. However, the crude extracts very often exhibited antibacterial activity [15,45,46]. The significant activity of *T. monadelpha* crude ethanol extract may be due to the synergic effect of its constituents.

#### 3.2.2. Antiplasmodial Activity

The antiplasmodial activity (Table 3) of the extract and some isolated compounds (**1–3**, **5**, **6**, **9**) against *Plasmodium falciparum* 3D7 (chloroquine sensitive/sulfadoxine resistant) was evaluated. Grandifotane A (**6**) and khayanolide D (**5**) showed good antiplasmodial activity with an IC_50_ of 1.37 µM and 1.68 µM, respectively, compared to the reference compounds chloroquine and artemisinin with IC_50_ values of 0.020 and 0.015, respectively. The mexicanolide-type limonoid grandifotane A (**6**) is several times more active than some previously reported compounds, such as 2,6-dihydroxyfissinolide (IC_50_ = 0.12 mM), fissinolide (IC_50_ = 48 µM), and 6-acetylsweitenolide (IC_50_ = 8.80–33.12 µg/mL) [47,48]. However, the antiplasmodial activity of polyoxyphragmalin or phragmalin had not yet been investigated [47,49]. Here, we have reported the first evaluation of the antiplasmodial activity of the phragmalin limonoids (**5**). Limonoids are highly oxygenated tetranortriterpenoids with different cyclization and substitution patterns. The activity exhibited by compounds **5** and **6** opens further questions in terms of the mechanism of action and the structure–activity relationship. In addition, this result indicated that the *Trichillia* genus, and possibly the Meliaceae family in general, may be a promising source of potential antiplasmodial secondary metabolites.

## 4. Conclusions

The present study was undertaken to investigate the chemical constituents of the stems bark of *T. monadelpha* which, to our knowledge, had not yet been studied for either their chemical composition or their biological activities. The chemical investigations undertaken on an ethanol extract of the stems bark of the plant led to the isolation and characterization of thirteen compounds, including two new limonoids, namely 24-acetoxy-21,25-dihydroxy-21,23-epoxytirucall-7-en-3-one (**1**) and (6*R*)-1-*O*-deacetylkhayanolide E (**2**). Extracts, sub-extracts, fractions, and some isolated compounds were assessed for their antibacterial and/or antiplasmodial activities on seven bacterial strains and on one plansmodium strain, respectively. Fraction E from the EtOAc sub-extract showed the best activity against *Staphylococcus aureus* ATCC 25923 (MIC of 3.9 µg/mL). Epicatechin (**9**) and melianodiol (**3**) showed significant and moderated activities against *Staphylococcus aureus* ATCC 25923 with the MIC of 7.8 µg/mL and 62.5 µg/mL, respectively, while grandifotane A (**6**) and khayanolide D (**5**) showed good antiplasmodial activities with IC_50_ values of 1.36 µM and 1.68 µM, respectively. Based on the above results, epicatechin and melianodiol could be the main antibacterial constituent of the stem bark of *Trichilia monadelpha*. As far as furanic limonoids are concerned, they were determined to be responsible for the antiplasmodial activity. For future re-investigations of the plant, collecting more starting material, evaluating its activity on other *Plasmodium falciparum* strains, as well as its toxicity and cytotoxicity, would all prove interesting areas of study.

## Figures and Tables

**Figure 1 metabolites-13-00298-f001:**
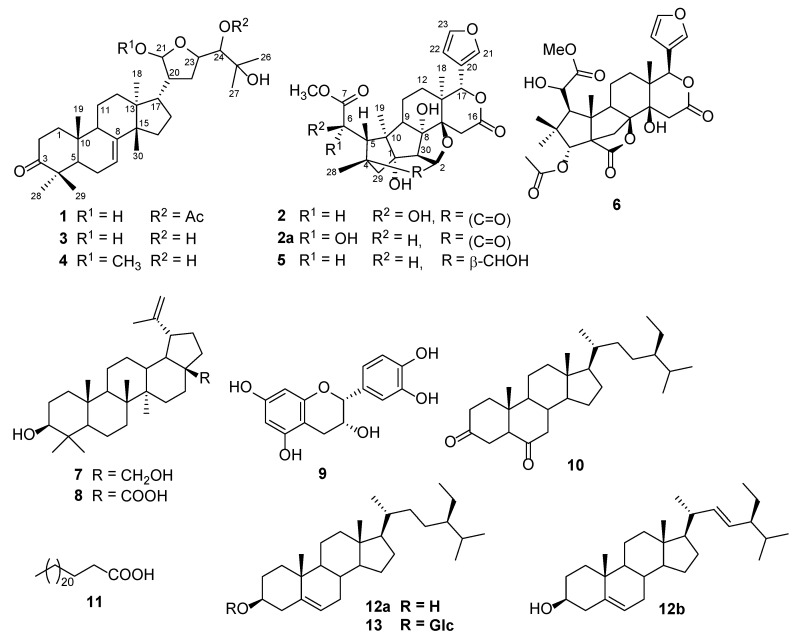
Structures of isolated compounds **1**–**13**.

**Figure 2 metabolites-13-00298-f002:**
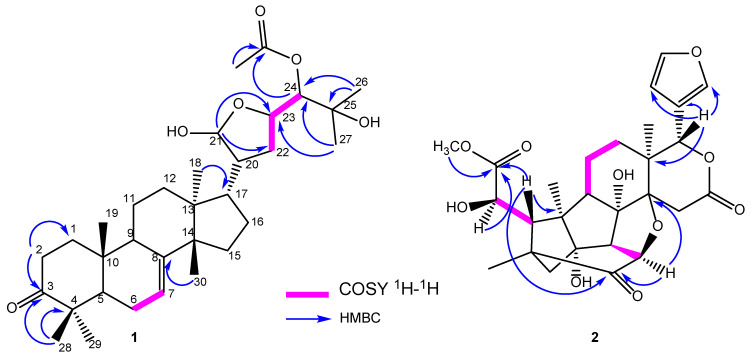
Important HMBC and ^1^H–^1^H COSY correlations in compounds **1** and **2**.

**Figure 3 metabolites-13-00298-f003:**
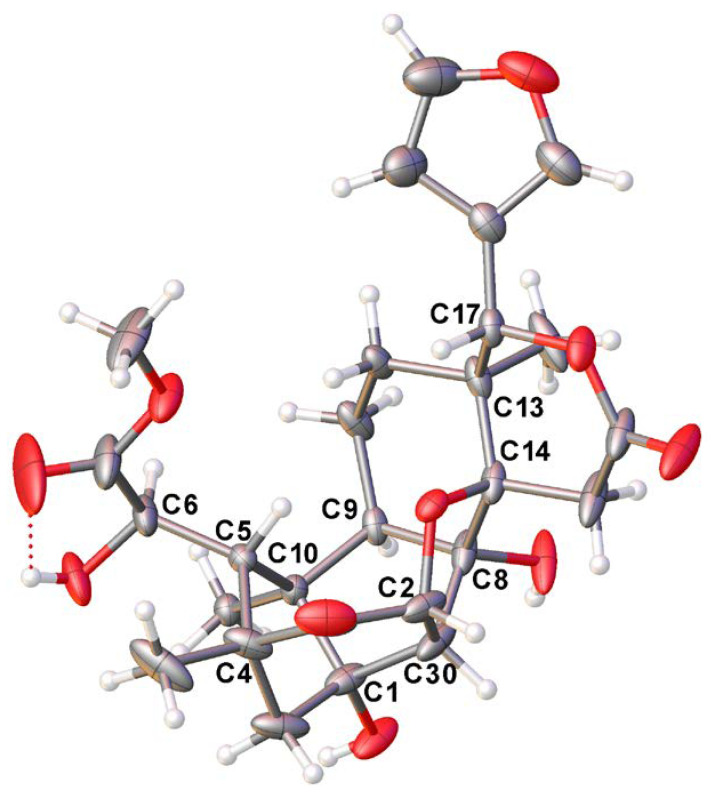
X-ray structure of compound 2; only the asymmetric carbon atoms are labeled.

**Table 1 metabolites-13-00298-t001:** ^1^H and ^13^C NMR data of compounds **1** and **2** (CD_3_OD, 500, and 125 MHz, respectively).

Position	1		2	
	δ_H_ (Integral, Multiplicity, J in Hz)	δ_C_	δ_H_ (Integral, Multiplicity, J in Hz)	δ_C_
1	2.06 (2H, m)	38.3/38.2		85.3
2	2.87/2.20 (1H, td, 14.5, 5.4)/(1H, m)	34.4/34.3	4.40 (1H, d, 10.5)	75.8
3		216.8		208.5
4		48.4		51.8
5	1.77 (1H, dd, 12.6, 6.9)	52.5/52.4	3.60 (1H, d, 1.7)	43.6
6	2.15 (2H, m)	24.0	4.49 (1H, d, 1.7)	72.9
7	5.37 (1H, brs)	117.9		176.5
8		145.7/145.8		88.5
9	2.13 (1H, m)	49.0	2.44 (1H, d, 9.5)	56.8
10		34.8		59.5
11	1.66 (1H, m)	17.5	1.77 (1H, m)/2.08 (1H, m)	17.1
12	1.81 (1H, m)	31.5/31.2	1.07 (1H, m)/1.94 (1H, m)	26.9
13		43.5/43.3		38.7
14		50.8/50.5		84.4
15	1.56 (2H, dd, 9.4, 2.3)	34.0/33.5	2.79 (1H, d, 16.9)/3.27 (1H, d, 18.9)	33.9
16	1.94 (1H, m)	27.1		173.0
17	2.11 (1H, m)	44.9	5.59 (1H, s)	82.1
18	0.90 (3H, s)	22.3	1.19 (3H, s)	14.8
19	1.08 (3H, s)	11.7	1.44 (3H, s)	20.4
20	1.97 (1H, m)	46.6		121.7
21	5.19/5.18 (1H, d, 5.3/3.6)	101.0/96.7	7.52 (1H, d, 2.6)	142.1
22	2.07 (2H, m)	31.0	6.46 (1H, d, 2.0)	110.8
23	4.25/4.34 (1H, dt, 14.5, 5.4)	77.3/75.4	7.53 (1H, t, 1.9)	144.0
24	4.99/4.92 (1H, d, 5.4)	79.6/78.9	-	-
25		71.2/71.1	-	-
26	0.93 (3H, s)	21.7	-	-
27	1.23 (3H, s)	25.2	-	-
28	1.05 (3H, s)	23.6	1.23 (3H, s)	15.7
29	1.15 (3H, s)	25.4/25.3	1.88 (1H, d, 12.5)/2.23 (1H, d, 12.6)	45.7
30	1.09 (3H, s)	26.4	2.91 (1H, d, 10.5)	64.8
1′		170.8		
2′	2.12 (3H, s)	19.7/19.6		
7-OMe			3.83 (3H, s)	53.0

**Table 2 metabolites-13-00298-t002:** Antibacterial activity of extracts, sub-extract, fractions, and some isolated compounds (MICs in µg/mL).

Extracts/Compounds	Minimum Inhibitory Concentrations (µg/mL)						
	ST	SA1	SA2	SA	PA	KP	KP1
EtOH extract	250	1000	1000	<62.5	125	1000	500
EtOAc sub-extract	>1000	500	1000	>1000	>1000	1000	500
*n*-BuOH sub-extract	>1000	>1000	>1000	>1000	>1000	>1000	>1000
FA	nd	>1000	>1000	nd	nd	>1000	>1000
FB	nd	>1000	>1000	nd	nd	>1000	250
FC	nd	>1000	>1000	nd	nd	>1000	>1000
FD	nd	>1000	>1000	nd	nd	>1000	>1000
FE	nd	3.9	250	nd	nd	1000	125
FF	nd	>1000	>1000	nd	nd	>1000	500
**1**	nd	>500	>500	nd	nd	>500	>500
**2**	nd	>500	>500	nd	nd	>500	>500
**3**	nd	62.5	125	nd	nd	500	62.5
**5**	nd	>500	>500	nd	nd	>500	>500
**6**	nd	>500	>500	nd	nd	>500	>500
**7**	nd	>500	>500	nd	nd	>500	>500
**9**	nd	7.8	125	nd	nd	500	500
Ciprofloxacin	0.015	0.0625	0.031	0.015	0.031	0.015	0.0625

**FA**: sub-fraction A; **FB**: sub-fraction B; **FC**: sub-fraction C; **FD**: sub-fraction D; **FE:** sub-fraction E; **FF**: sub-fraction F; **ST**: *Salmonella typhi*; **SA1**: *Staphylococcus aureus* ATCC 25923; **SA2**: *Staphylococcus aureus* clinical isolate; **SA**: *Staphylococcus aureus* ATCC 43300; **PA**: *Pseudomonas aeruginosa* NR 48982; **KP**: *Klebsiella pneumoniae* NR 41388; **KP1**: *Klebsiella pneumoniae* clinical isolate; **nd**: not determined.

**Table 3 metabolites-13-00298-t003:** Antiplasmodial activity of extracts and some isolated compounds.

Extracts and Compounds	% Growth Inhibition	IC_50_ (µM)
EtOH extract	0	nd
1	24.3	nd
2	7.69	nd
3	25.48	nd
5	45.07	1.68
6	54.17	1.37
9	34.87	nd
Chloroquine	nd	0.020
Artemisinin	nd	0.015

**nd**: not determined.

## Data Availability

Data are available in the Appendix A.

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
