# Peer review of "Constituents of the Stem Bark of Trichilia monadelpha (Thonn.) J. J. De Wilde (Meliaceae) and Their Antibacterial and Antiplasmodial Activities"

_metabolites, 2023, doi:10.3390/metabo13020298_

Round 1

Reviewer 1 Report

The manuscript entitled “Constituents of the stem bark of Trichilia monadelpha (Thonn.) J. J. De Wilde (Meliaceae) and their antibacterial and antiplasmodial activitiesreported the isolation and structural elucidation of two new limonoids 24-acetoxy-21,25-dihydroxy-21,23-24-epoxytirucall-7-en-3-one (1) and (6R)-1-O-deacetylkhayanolide E (2), together with eleven known compounds (3-13). Regarding antibacterial activity, they found that  fraction E showed significant activity against S. aureus with an MIC value of 3.90 µg/mL, while epicatechin (9) exhibited significant activity with MIC values of 7.80 µg/mL.

In addition, khayanolide D (5) and grandifotane A (6) showed antiplasmodial activities against the chloroquine-sensitive/sulfadoxine-resistant plasmodium falciparum 3D7 strain with IC50 values of 1.68 µM and 1.37 µM, respectively. However, the corresponding plant extract and fractions did not exhibited antiplasmodial activities.

In my opinion, the manuscript is not good enough for publication in Metabolites journal. The author got 2 new compounds but compound 1 was a mixture, the authors could not assigned stereo centers at C-21, C-23 and C-24. Compounds 7-13 are very common metabolites in the plant.

1.      Why the author did not evaluate the biological activity of compound 2 (antibacterial acitivity) and 4.

2.      Correct the label in the figure 1.

3.      Correct HRMS data of compound 2 in the manuscript, not match with SI

4.      It seems that HRMS obtained from 2 different mass spectrometers. Please add in the 2.1. part.

5.      In the SI, the 1H-NMR spectrum did not show integrations of peaks.

Reviewer 2 Report

Dear Authors.

The present study was undertaken to investigate the chemical constituents of the stem bark of T. monadelpha which, in our knowledge was not yet studied for their chemical composition as well as for their biological activities. The chemical investigations undertaken on ethanol extract of the stems bark of the plant led to the isolation and characterization of fortune compounds, including two new limonoids: 24-acetoxy-21,25-di-hydroxy-21,23-epoxytirucall-7-en-3-one(1) and (6R)-1-O-deacetylkhayanolide E (2).

Your Article made a very good impression, the article is written in a good understandable language and preliminary scientific conclusions will be informative. The plant T. monadelpha exhibits so many interesting biomedical activities that it simply needs to be studied in great detail, going far beyond the scope of this study, I urge you not to abandon the study of this interesting biological and biotechnological object.

At the same time, I have a number of small comments on your article in order for you to expand the discussion somewhat.

1. You write:

240 In general, the antibacterial activity of limonoids is very poor. However, the crude e[tracts very often exhibit antibacterial activity [15, 43, 44].

Is it possible to explain this statement in more detail? Perhaps other authors used more efficient extraction methods or experimental protocol, or did some other factors intervene? This is an extremely interesting idea.

2. You write:

257 However, the antiplasmoidal activity of phragmalin had not yet been investigated. We reported here the first evaluation of the antiplasmoidal activity of the phragmalin limonoids.

This achievement deserves a wider comment, especially since you show the result achieved for the first time.

Best regards.

Reviewer 3 Report

Regarding the manuscript entitled "Constituents of the stem bark of Trichilia monadelpha (Thonn.) J. J. De Wilde (Meliaceae) and their antibacterial and antiplasmodial activities" I have some minor comments to improve its quality. In general, in my opinion the main significant results is isolation and structural identification of two novel limonoids. Thus I highly recommend strengthen the introduction and discussion part. 

Introduction

please bring the main phytochemicals identified from the species, just classifications is not enough.

L64-65: Please check the English language of the whole MS text: e.g. "The TLC plate were visualized"

Results 

Please describe the reason for using the EtOAc soluble fraction? have you bio-assayed before subjecting to separation procedure? any evidence in literature?

Discussion part is too weak, short, and superficial. I highly recommend adding more in details comparing the similar findings.

Conclusion

this section is just summary of the results, please improve it with mentioning perspectives, suggesting further studies

Round 2

Reviewer 1 Report

The authors have corrected the manuscript to improve its quality.

 The manuscript can be accepted after minor revisions.

1. Pgae 5, line 174: Change "and" to "an"

2. Line 189, 208, 220: 1H-1H COSY, there is a space.

3. Figure 3: Numbered carbon atoms are not matched with Figure 1. Re-number again.

4. Line 296: Epicatechin 

Reviewer 3 Report

Authors have revised the manuscript in accordance with the comments suggested and items requested.
